# Sex- and Gender-Based Analysis on Norepinephrine Use in Septic Shock: Why Is It Still a Male World?

**DOI:** 10.3390/microorganisms12040821

**Published:** 2024-04-18

**Authors:** Benedetta Perna, Valeria Raparelli, Federica Tordo Caprioli, Oana Teodora Blanaru, Cecilia Malacarne, Cecilia Crosetti, Andrea Portoraro, Alex Zanotto, Francesco Maria Strocchi, Alessandro Rapino, Anna Costanzini, Martina Maritati, Roberto Lazzari, Michele Domenico Spampinato, Carlo Contini, Roberto De Giorgio, Matteo Guarino

**Affiliations:** 1Department of Translational Medicine, University Hospital of Ferrara, 44124 Ferrara, Italy; prnbdt@unife.it (B.P.); federi.tordocaprioli@edu.unife.it (F.T.C.); oanateodora.blanaru@unife.it (O.T.B.); cecilia.malacarne@unife.it (C.M.); cecilia.crosetti@unife.it (C.C.); andrea.portoraro@unife.it (A.P.); alex.zanotto@unife.it (A.Z.); francescomaria.strocchi@unife.it (F.M.S.); alessandro.rapino@unife.it (A.R.); anna.costanzini@unife.it (A.C.); spmmhl@unife.it (M.D.S.); grnmtt@unife.it (M.G.); 2Department of Translational and Precision Medicine, Sapienza University of Rome, 00161 Rome, Italy; valeria.raparelli@uniroma1.it; 3Infectious Diseases Unit, Department of Medical Sciences, University of Ferrara, 44121 Ferrara, Italy; martina.maritati@unife.it (M.M.); cnc@unife.it (C.C.); 4Emergency Department, Hospital de la Santa Creu I Sant Pau, 08041 Barcelona, Spain; rlazzari@santpau.cat; 5Emergency Department, University Hospital of Ferrara, 44124 Ferrara, Italy

**Keywords:** sex, gender, septic shock, norepinephrine, vasoactive agents, precision medicine

## Abstract

Sex and gender are fundamental health determinants and their role as modifiers of treatment response is increasingly recognized. Norepinephrine is a cornerstone of septic shock management and its use is based on the highest level of evidence compared to dopamine. The related 2021 Surviving Sepsis Campaign (SCC) recommendation is presumably applicable to both females and males; however, a sex- and gender-based analysis is lacking, thus not allowing generalizable conclusions. This paper was aimed at exploring whether sex- and gender-disaggregated data are available in the evidence supporting this recommendation. For all the studies underpinning it, four pairs of authors, including a woman and a man, extracted data concerning sex and gender, according to the Sex and Gender Equity in Research guidelines. Nine manuscripts were included with an overall population of 2126 patients, of which 43.2% were females. No sex analysis was performed and gender was never reported. In conclusion, the present manuscript highlighted that the clinical studies underlying the SCC recommendation of NE administration in septic shock have neglected the likely role of sex and gender as modifiers of treatment response, thus missing the opportunity of sex- and gender-specific guidelines.

## 1. Introduction

Sepsis is a life-threating organ dysfunction caused by a dysregulated host response to infection [1,2]. This time-dependent condition affects over 2 million people/year worldwide with high mortality burden steadily growing in the last years [3,4]. Furthermore, sepsis incidence has constantly increased since the first consensus definition (Sepsis-1) in 1991, reaching around 49 million cases and 11 million sepsis-related deaths worldwide in 2017 [3,4]. The increase in its incidence is a matter of immediate concern and the World Health Organization (WHO) has recently declared sepsis as a global health priority [4]. According to the third consensus expressed by the Surviving Sepsis Campaign (SSC), sepsis is diagnosed in case of an increase in Sequential Organ Failure Assessment (SOFA) score ≥ 2 from baseline. Septic shock is a subset of sepsis defined by the need of a vasopressor to maintain a mean arterial pressure (MAP) ≥ 65 mm Hg and serum lactate level ≥ 2 mmol/L [1].

Sepsis (and in particular septic shock) is burdened by a severe vasoplegia which can lead to a distributive shock [5,6]. Therefore, an effective support to hemodynamics is crucial for patients’ survival [2]. Despite significant advancements, the circulatory support remains challenging and it is based on two main pillars: (i) fluids (mainly crystalloids) and (ii) vasoactive agents [1,2]. In particular, the use of the latter represents one of the main cornerstones of septic shock treatment [2,7]. Among vasopressors, SSC strongly recommended norepinephrine (NE) as first-choice drug, with high quality of evidence as compared to dopamine (DA) [2]. NE is an α-1/β-1 adrenergic agonist which predominantly affects the peripheral vascular system. At this level, it enhances vascular filling pressure and redistributes blood flow by its venoconstrictive effect [8]. Furthermore, NE improves coronary perfusion, myocardial contractility, and cardiac output with a minor impact on heart rate [9,10,11,12,13,14,15,16,17,18]. DA also affects both alpha-adrenergic and beta-adrenergic receptors, with higher affinity to the latter [19]. This vasopressor works as a dose-dependent agent on dopamine-1, α-1, and β-1 adrenergic receptors. Indeed, at lower dosages, DA causes vasodilation in the renal, splanchnic, cerebral, and coronary circulation. Contrariwise, at higher dosages, its predominant effect is represented by vasoconstriction and the subsequent increase in systemic vascular resistance [2]. Moreover, its β-1 adrenergic receptor activity can lead to cardiac arrhythmias, limiting its use [2]. In a recent systematic review and network meta-analysis on thirty-three randomized controlled trials (RCTs) comprising 4966 patients, DA was associated with the highest incidence of cardiac arrhythmia and a higher risk of 28-day mortality due to septic shock compared to the other vasopressors [20].

Sex and gender are fundamental health determinants and they contribute to variations in clinical outcomes and response to treatment [21,22,23,24]. The term “sex” describes a set of biological attributes related to physical, genetic, and physiological features (i.e., chromosomes, gene expression, hormone function, and reproductive/sexual anatomy) and is usually categorized as female or male. Instead, the term “gender” expresses the socially constructed roles, behaviors, and identities of women, men, and gender-diverse people and it is usually superficially conceptualized as binary (woman/man) [21,22]. The main factors affecting sex and gender are highlighted in Figure 1.

Despite the increasing awareness about investigating sex- and gender-dependent differences in response to treatments, females were usually underrepresented in clinical studies assessing the efficacy and safety of interventions, as extensively reported in the context of cardiovascular (CV) drugs [25]. Given the absence of other confounding variables, the relationship between biological sex and treatment response could be properly evaluated in animal models. However, a sex bias in preclinical research was reported, secondary to the selection of a disproportionately high number of male animals in many investigational areas of biology [26]. In sepsis, an alarming paucity of data exploring sex-differences in response to treatments was demonstrated both in septic animals [27] and humans [28,29]. Moreover, in human patients, sepsis bundles were fulfilled more often in males compared to females and, inversely, median time to antibiotic administration was longer in females compared to the counterpart, reflecting treatment disparities between the two sexes [29]. Compared to sex, gender is even less frequently evaluated in clinical studies. This evidence is probably related to gender multidimensional aspects [30]. Indeed, the relationship between gender and sepsis management is poorly described [31]. However, it has been previously reported how both sex and gender play a pivotal role in sepsis: in particular, women seem to have a less reactive inflammatory response and to recover more effectively than the counterpart, despite receiving less invasive therapies [32]. Specifically, males affected by septic shock are more frequently admitted in Intensive Care Units (ICUs) compared to females [30]. Sex- and gender-related differences have also been described in vascular response to adrenergic stimulation and they might be related to the responsiveness of vascular β-adrenergic receptors to catecholamine stimulation, which is higher in young women compared to age-matched men [33,34,35,36,37,38,39]. Moreover, the vascular expression of β-adrenergic receptors is mainly related to estrogens, which explains the reduced responsiveness observed in aging in women. This reduction leads to an unopposed α-adrenergic vasoconstriction, a rise in blood pressure and an increase in cardiovascular risk [33,37,40,41]. Nevertheless, the SCC recommendations on sepsis treatments are commonly sex blinded.

Based on this background, with the present study we aimed at exploring whether sex- and gender-disaggregated data are available in the evidence supporting the guidelines’ recommendation about the use of NE vs. DA. We questioned the “one size fits for all” approach in septic patients unless adequately proven, since, compared to DA, NE use is the only treatment supported by the highest level of evidence.

## 2. Materials and Methods

All the RCTs or prospective studies quoted for supporting the 2021 SSC recommendation on the use of NE vs. DA in septic shock were retrieved.

Exclusion criteria were (i) full text not retrievable; (ii) language different than English; and (iii) no data about sample size.

For each manuscript, the following information were extracted: (i) first author name; (ii) year of publication; (iii) study design (i.e., multicentric); (iv) sample size; (v) primary outcome; (vi) conclusions. According to the Sex and Gender Equity in Research (SAGER) guidelines [42], the subsequent data were abstracted: reported identity (if sex or gender); presence of gender-diverse people (e.g., transgender people); numerosity of female; numerosity of male; numerosity of gender-diverse people; sensitivity analysis by sex; subgroup analysis by sex; post-hoc analysis by sex; disaggregated report of results by sex; sensitivity analysis by gender; subgroup analysis by gender; post-hoc analysis by gender; disaggregated report of results by gender.

Four pairs including a woman and a man (FTC and FMS, TOB and AR, CM and AZ, CC and AP), independently extracted the data using a standardized data abstraction form (Excel spreadsheet). Disagreements were resolved by adjudication with another investigator (BP).

All statistical analyses were conducted with Statistical Product and Service Solution (SPSS) 23.0 for Windows (IBM Corp., Armonk, NY, USA). Categorical data are expressed as absolute frequencies and percentages, while medians and interquartile range (IQR) are reported for continuous variables.

## 3. Results

Nine eligible manuscripts were identified [43,44,45,46,47,48,49,50,51]. The main features of the selected studies are summarized in Table 1 and Table 2. Manuscripts were published from 1989 to 2011 and included seven RCTs and two prospective studies, none was multicentric and the majority enrolled less than 50 individuals. The overall population included 2126 patients; among them, 919 (43.2%) were female. In three out of nine manuscripts, NE was considered more effective and reliable than DA [43,44,48]; in the remaining, the efficacy of the two agents was considered comparable but DA was associated with more side effects (even life-threatening) than NE [45,46,47,49,50,51]. In 21 (1%) patients derived from the analysis of Ruokonen et al. [45], sex was not even specified. Gender was not reported in any of the included manuscripts; consequently, gender-based analysis was not feasible. In seven out of nine studies [44,45,46,47,48,49,51], females represented far less than 50% of the included population. No sex-based analysis was performed in any of the included manuscripts; thus, the results were never reported as disaggregated by sex.

## 4. Discussion

The present analysis revealed that clinical studies testing the efficacy and safety of NE for septic shock have neglected the likely role of biological sex and sociocultural gender as modifiers of treatment response. Notably, none of the studies supporting the 2021 SCC recommendation on the use of NE compared to DA reported sex-disaggregated findings. Therefore, the opportunity of sex-specific guidelines is missed, with potential harm for patients with sepsis. These results appeared counterintuitive as there is increasing, albeit still scarce, evidence on sex-based differences in the epidemiology, clinical presentation, management, and outcome of patients with sepsis. Notably, conflicting data have been reported about sepsis incidence. Indeed, several nationwide studies reported a higher risk of sepsis in males compared to females [32], although latest data reported a higher age-standardized incidence of sepsis in women [52]. Specifically, septic males appeared more prone to be admitted to ICU [53] and to develop septic shock within 48 h from the admission [54]. Several explanations could be theorized for this finding, including gene polymorphisms [55] and sex-dependent differences in the immune response (e.g., higher levels of proinflammatory cytokines and procalcitonin in male septic patients) [27], although the related mechanisms are still poorly understood. Further sex-related differences were described in clinical presentation of septic patients; indeed, females are at higher risk of developing urinary trait infections [56], while males are more prone to endocarditis and respiratory and mycotic infections [32]. Moreover, male sex has been associated with a higher incidence of sepsis-induced cardiomyopathy compared to the counterpart [57]. This phenomenon is presumably due to the role of estrogen in modulating several injury-related myocardial responses, especially at a cellular level [32]. Regarding the relationship between sex and sepsis-related mortality, findings are conflicting, inconclusive [32,53,54] and still a matter of debate [58].

Although unraveling sex-differences in septic patients could lead to a more personalized management of these subjects, Antequera et al. highlighted a female under-representation in sepsis studies [59]. Indeed, in over 200 manuscripts analyzed by these authors, the mean percentage of enrolled females reached 40% of the overall population, a result consistent with our findings. Low participation of women among enrolled patients is quite common also in trials dealing with other diseases, e.g., heart failure (HF), coronary artery disease, and acute coronary syndrome [60]. Under-enrollment in those studies has been attributed to different reasons, including the presence of comorbidities [61] and male-patterned inclusion criteria (e.g., in HF trials, a cut-off value <40% of ejection fraction almost consistently excluded or markedly reduced the number of recruited women, known to be predominantly affected by HF with preserved ejection fraction—HFpEF) [25,62]. This lack of female inclusion in clinical studies has been associated with suboptimal medical management in this population [63]. Several factors related to the absence of systematic facilities (e.g., lack of incentives), trial design (e.g., exclusion of pregnant women), and participants’ characteristics (e.g., childcare and eldercare responsibilities) were described as barriers to women’s participation in clinical research [64]. To bridge this gap, several institutions have attempted to ensure equity in the participation to clinical trials [63]. In 1994, the National Institute of Health (NIH) published the guidelines on the inclusion of women and minorities in clinical research [65]. This document, amended in 2017, was intended to warrant that all NIH-supported research involving human subjects would be performed in order to obtain systematic data about individuals of both genders and the diverse racial/ethnic groups. Specifically, in the case of RCTs, the NIH recommended to investigate the effect of any intervention on such groups. Moreover, in 2016 a panel of experts developed the SAGER guidelines [42], a comprehensive tool for standardizing and reporting data about sex and gender in study design, statistical analyses, results, and interpretation of findings. These guidelines provided editors with a tool able to evaluate the presence of SGBA in submitted manuscripts and eventually promote this analysis.

Since 2013, the females enrolled were more than one-half of all subjects in NIH studies on human subjects [66]. Notably, most investigators have focused on sex matching, pairing females with males to warrant equity and adequate representation of both groups [63]. Nevertheless, sex bias persists in human research, with male and female subjects underrepresented in different fields. In particular, a significative female under-enrollment was demonstrated in the clinical trials pertaining to the disciplines of infectious diseases and cardiology [63]. Furthermore, several organizations, including the NIH, are advocating the inclusion of sex-based considerations and analyses in preclinical studies in vitro on cell cultures and in vivo in experimental animals. These efforts may turn into significant key differences among groups that could guide clinical research and contribute to establish the reproducibility of preclinical research [66]. The same rigorous endeavor to produce data on men’s and women’s health was not attempted for diverse-gender people. Indeed, a small number of clinical studies reported data on non-binary or transgender people [63]. The lack of data on gender-diverse people limits the generalizability of human research to these communities and advocates a real inclusion of these groups in clinical studies. To further bridge the gap in the representation of both sexes in all fields of research, SAGER guidelines provide a comprehensive tool for reporting information about sex and gender in study design, data analyses, results, and interpretation of findings. Inter alia, if sex and gender are considered relevant to the topic of the study, it is recommended that authors should report disaggregated data by sex and gender and consider SGBA, or lack thereof, in the discussion and among limitations of their manuscript. Otherwise, a justification about the reasons why sex and gender do not affect the outcome of the study should be provided.

Despite the SSC guidelines having been published in 2021, no data on sex and/or gender were reported in any recommendation. Beyond the complete absence of a sex- and gender-based analysis (SGBA), the authors did not attempt an explanation for this lack nor any hypothesis of future interest in this field. In particular, although sex-specific differences in vascular receptors were reported [67], neither in the recommendation of NE administration in septic shock nor in the studies underlying this indication, there are data on sex and/or gender. The latter finding is understandable, considering that these manuscripts were published before 2016; however, this explanation is not acceptable for the latest SSC guidelines.

Furthermore, our analysis showed that none of the eligible manuscripts reported sex- or gender-disaggregated data. This finding is in line with the analysis of Antequera et al., who highlighted that only 57 out of 277 included studies proposed disaggregated analyses [59]. Investigators should report data by sex/gender as well as test the interactions between these factors and the main outcomes to obtain women-specific results. The inadequate representation of women in research could result in several significant issues, including sex-biased outcomes of measurements [25], and limit the generalizability of research findings and their applicability to clinical practice.

In the included manuscripts, gender was never mentioned and the lack of this analysis was not even discussed. Since gender, as well as sex, is a critical determinant of health, specific analyses should be performed including gender-diverse people.

Finally, the contribution of sex and gender to individual and global health is crucial, but it is mandatory to consider them in a broader dimension, which includes intertwined factors, such as ethnicity, religion, educational level, and socio-economic status. Intersectionality represents a multidimensional approach useful to understand how these interlaced elements interact to shape our identity, behaviors, and complex interpersonal relationships [68]. An intersectional analysis is progressively taking hold also in clinical research, particularly tackling areas such as addictions [69], neurological disorders [70], or infectious diseases [71].

Systematic female under-representation in sepsis studies led to unexplored sex differences in sepsis host response and limitation to clinical translation [72]. Therefore, this manuscript questioned the generalizability of the strong recommendation for NE administration in septic shock and advocated for methodical SGBA. This study shows different limitations: (i) the inadequacy of the involved literature, even though underpinning the latest SSC recommendation on vasopressors; (ii) the limited number of eligible studies; and (iii) the small sample size of some of the included manuscripts.

## 5. Conclusions

The present manuscript emphasized the complete absence of sex and gender reported data in the studies underpinning the latest SSC recommendations for the use of NE in septic shock. This finding is in contrast with the SAGER guidelines, which aimed at improving sex and gender reporting in scientific research. Since different manuscripts demonstrated sex- and/or gender-based differences in preclinical and clinical investigations, the scientific community should be aware of the fundamental role of SGBA in enhancing the quality of research.

## Figures and Tables

**Figure 1 microorganisms-12-00821-f001:**
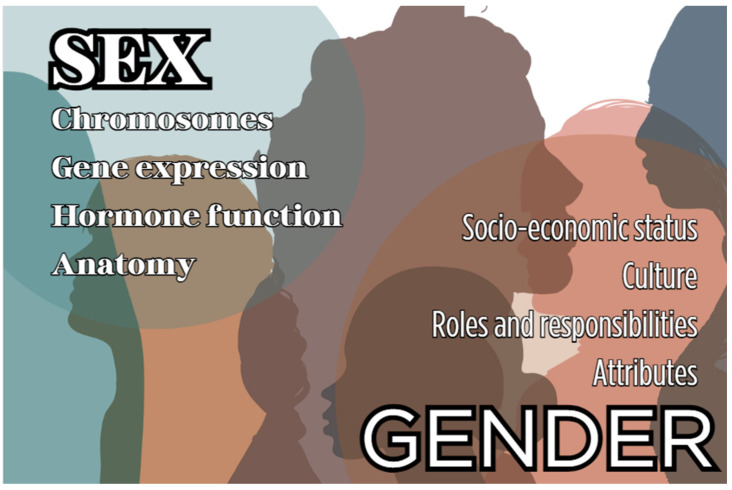
Main factors affecting sex and gender.

**Table 1 microorganisms-12-00821-t001:** Synopsis highlighting the main features of the involved studies.

First Author	Year	Study Design	Multi-Centric	Patients (n)	Mean Age (y)	Primary Outcome	Conclusions
Schreuder et al. [43]	1989	Prospective study	No	10	57	Compare NE vs. DA on hemodynamics, oxygen metabolism, and right ventricular performance.	NE was superior to DA in all the outcomes.
Martin et al. [44]	1993	RCT	No	32	52	Compare NE vs. DA in reversing hemodynamic and metabolic abnormalities in septic shock.	NE is more effective and reliable than DA.
Ruokonen et al. [45]	1993	RCT	No	21	43	Compare NE vs. DA in measuring the blood flow distribution and regional oxygen transport in septic shock.	Both NE and DE improved blood flow distribution and oxygen transport in septic shock.
Marik et al. [46]	1994	RCT	No	20	46	Compare NE vs. DA on systemic and splanchnic hemodynamics in septic patients.	DA may cause an uncompensated increase in splanchnic oxygen requirement in septic patients. NE may have a more favorable hemodynamic profile.
Guerin et al. [47]	2005	Prospective study	No	12	40	Compare NE vs. DA on systemic and splanchnic hemodynamics in septic patients.	NE was as effective as DA in maintaining splanchnic perfusion. The metabolic response might favor NE.
Mathur et al. [48]	2007	RCT	No	50	53	Compare NE vs. DA in reversing hemodynamic and metabolic abnormalities in sepsis.	NE is more effective and reliable than DA on primary outcome.
De Backer et al. [49]	2010	RCT	Yes	1679	67	Compare NE vs. DA on 28-day mortality in patients with shock.	No differences between NE and DA were detected. DA was associated with more adverse events.
Patel et al. [50]	2010	RCT	No	252	N/A	Compare NE vs. DA on 28-day mortality in septic shock.	No differences between NE and DA were detected. DA was associated with more adverse events.
Agrawal et al. [51]	2011	RCT	No	50	53	Compare NE vs. DA in reversing hemodynamic and metabolic abnormalities in septic shock.	NE was more useful than DA in reversing hemodynamic and metabolic abnormalities in septic shock.

**Table 2 microorganisms-12-00821-t002:** Summary of the sex- and gender-based analysis.

Authors	Reported Identity (Sex/Gender)	Other Categories (Yes/No)	Females n (%)	Males n (%)	Sensitivity Analysis by Sex (Yes/No)	Subgroup Analysis by Sex (Yes/No)	Post-Hoc Analysis by Sex (Yes/No)	Disaggregated Report of Results by Sex (Yes/No)
Schreuder et al. [43]	Sex	No	6 (60.0%)	4 (40.0%)	No	No	No	No
Martin et al. [44]	Sex	No	8 (25.0%)	24 (75.0%)	No	No	No	No
Ruokonen et al. [45]	N/A	No	N/A	N/A	No	No	No	No
Marik et al. [46]	Sex	No	9 (45.0%)	11 (55.0%)	No	No	No	No
Guerin et al. [47]	Sex	No	1 (8.3%)	11 (91.7%)	No	No	No	No
Mathur et al. [48]	Sex	No	18 (36.0%)	32 (64.0%)	No	No	No	No
De Backer et al. [49]	Sex	No	723 (43.1%)	956 (56.9%)	No	No	No	No
Patel et al. [50]	Sex	No	136 (54.0%)	116 (46.0)	No	No	No	No
Agrawal et al. [51]	Sex	No	18 (36.0%)	32 (64.0%)	No	No	No	No

Note: DA: dopamine; N/A: not available; NE: norepinephrine; RCT: randomized controlled trial.

## Data Availability

No new data were created or analyzed in this study. Data sharing is not applicable to this article.

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
