# Peer review of "Sex- and Gender-Based Analysis on Norepinephrine Use in Septic Shock: Why Is It Still a Male World?"

_microorganisms, 2024, doi:10.3390/microorganisms12040821_

Round 1

Reviewer 1 Report

Comments and Suggestions for Authors

I reviewed the article entitled “Sex- and gender-based analysis on norepinephrine use in septic shock: why is it still a male world?” by Perna et al. submitted to Microorganisms (Manuscript ID: microorganisms-2857704). In this study, the authors found that the lacks of a sex and gender-based analysis in current study. From their observations, they claimed that the need of sex-and gender- specific sepsis treatment guidelines.

First, the reviewer pays respect for the Authors' effort spent on this manuscript. However, there are numerous concerns which are listed below:

First, all results and discussions are based on the previous study, and not the author's own data. This means almost all contents of this manuscript is not based on scientific efforts. This manuscript therefore does not reach the high level of the original articles included in this journal.

Even if this manuscript is the systematic review, this manuscript does not conform the PRISMA Statement (http://www.prisma-statement.org/) Please respect the basic rule of scientific writing.

In addition, most of the discussion section is simply list the findings from previous study, or rephrase the results. This reviewer thinks the discussion is also not thought evoking one.

There are many other comments, but this reviewer cannot list them all in the limited review time. These flaws are unlikely to be correctable through any amount of peer review comments.

Although the criticisms listed above, the reviewer respects the authors’ time and effort spent on this manuscript, and the authors ‘patience and professionalism in dealing with my comments.

Comments on the Quality of English Language

The reviewer rocommends to use the professional English editting survice.

Reviewer 2 Report

Comments and Suggestions for Authors

This study entitled " Sex- and gender-based analysis on norepinephrine use in septic shock: why is it still a male world?" by Perna B. et al. was designed to explore whether sex-and gender-disaggregated data are available in the evidence supporting the Surviving Sepsis Campaign (SCC) recommendation. The inclusion of 2126 patients and the explicit gender-related data extraction process add to the robustness of the findings. Of the nine manuscripts analyzed, none provided gender analysis or reported gender information, a striking finding that reveals an important aspect of the existing literature that has been overlooked.

Summary,

The introduction clearly presents the context of the research model and articulates the research questions and objectives.

The results section is clear, well-organized, and descriptively robust.

The discussion and conclusions are correct and appropriately compared with the literature.

Reviewer 3 Report

Comments and Suggestions for Authors

First of all I am happy to congratulate for the originality and the quality of the study. Actually, the lack of informations about the sex and gender-related data  are clear as well as the possible clinical consequences. The manusctipt could be fuither improved by adding some hypotheses about the causes of this absence of informations.

Reviewer 4 Report

Comments and Suggestions for Authors

This is an important subject and I agree, women are underrepresented in the literature. However the authors have decided to base their arguments on old literature, comparing NA with dopamine. These papers date from 1989 to 2011. Although the analysis is adequate, there is no consideration of updated literature and it feels as if the comparison was chosen in order to make an already held point rather than an investigation of the literature. 

I have included some relevant literature below

The mean age of the participants in chosen studies was very young for most ICUs, All bar one study had mean ages of < 60 and in the exception the mean age was 67. 

It would be good to see more hypothesis about why sex should make and difference, some were included, but not all. For example women may be more prone to cardiac dysrhythmias than men who suffer from coronary ischaemia more often.

The text is somewhat conflicting. in the introduction it is stated that women are less susceptible to sepsis and recover more effectively than men, despite receiving less invasive therapies. This is highlighted in other studies and the 43% of women, is likely to represent an appropriate proportion without any conspiracy. In the discussion you point out that globally (Rudd et AL) more females develop sepsis but you neglect to point out that the same study suggests men are more likely to die, implying that their condition is more severe. This global study is also not restricted to ICU patients. 

I agree that Gender analysis is lacking and may be important. It would be useful to understand what factors the authors consider important. Gender is a social construct and very variable. One person's gender reassignment may simply be a choice of identity and for others medical intervention including hormonal therapy and surgery are included. If there are differences based on gender, the choices made are likely to be important.  

Including gender to any important level, would require very large studies to include sufficient numbers of patients.

If the authors consider differences to be caused by bias, this has to be evidenced more carefully. Using the old literature is not very helpful. 

Thompson KJ, Finfer SR, Woodward M, Leong RNF, Liu B. Sex differences in sepsis hospitalisations and outcomes in older women and men: A prospective cohort study. J Infect. 2022 Jun;84(6):770-776. doi: 10.1016/j.jinf.2022.04.035. Epub 2022 Apr 25. PMID: 35472366.

Lamontagne F, Richards-Belle A, Thomas K, Harrison DA, Sadique MZ, Grieve RD, Camsooksai J, Darnell R, Gordon AC, Henry D, Hudson N, Mason AJ, Saull M, Whitman C, Young JD, Rowan KM, Mouncey PR; 65 trial investigators. Effect of Reduced Exposure to Vasopressors on 90-Day Mortality in Older Critically Ill Patients With Vasodilatory Hypotension: A Randomized Clinical Trial. JAMA. 2020 Mar 10;323(10):938-949. doi: 10.1001/jama.2020.0930. PMID: 32049269; PMCID: PMC7064880.

Guidet B, Maury E. Sex and severe sepsis. Crit Care. 2013 May 15;17(3):144. doi: 10.1186/cc12690. PMID: 23680409; PMCID: PMC3672659.

Sakr Y, Elia C, Mascia L, Barberis B, Cardellino S, Livigni S, Fiore G, Filippini C, Ranieri VM. The influence of gender on the epidemiology of and outcome from severe sepsis. Crit Care. 2013 Mar 18;17(2):R50. doi: 10.1186/cc12570. PMID: 23506971; PMCID: PMC3733421.

Round 2

Reviewer 1 Report

Comments and Suggestions for Authors

The authors just modified the text, and have not addressed any of my comments scientifically. The problems lie in the design and methodology, and such flaws are unlikely to be correctable through peer review.
